

# Analyzing reciprocity dynamics in supply chains of public goods: a stochastic evolutionary game approach

Simo Sun[1], Man Wang[2] and Yi Lei[3]

[1] School of Mathematics and Statistics, Guizhou University of Finance and Economics, Guiyang, China
[2] School of Management Science, Guizhou University of Finance and Economics, Guiyang, China
[3] Guizhou Leading Network Technology Services Co. Ltd, Guiyang, China

## ABSTRACT

To start with an infinitely repeated game of supply chains of public goods, a stout reciprocity mechanism is introduced into income games to build a matric dynamic equation. The conventional evolutionary game method is employed to propose a model called the evolutionary game for the cooperative strategy of both the manufacturer and the seller groups in the supply chain of public goods. Also, white Gaussian noise (WGN) is added to reflect random interference in the evolution process. Then, a stochastic dynamic system is established, and Ito's differential equation is used to analyze both sides' strategy evolution in a game, interpret changes in system stability when random disturbance is added, and finally test the influence of different situations on the system stability by running a numerical simulation. The research shows that the stronger the reciprocity coefficient is, and the system is subjected to random interference, the greater the strategy choice change in players' decision-making procedures when the repeated game of public goods is conducted.

## INTRODUCTION

Public goods are a sort of commodity. In The Pure Theory of Public Expenditure, published in 1954, (*Samuelson, 1954*) an American economist, indicated that public goods are products and services that several people utilize in the same period. Public goods do not have rivals when they are consumed or used and are not excludable in advantage.

A large amount of data on public goods in experimental games, as outlined by *Cardenas & Carpenter (2008)*, confirms that complete rationality in human behavior is not a strong assumption. *Chen, Ye & Wang (2012)* indicated that individuals do not show complete rationality and choose one of the options when preferences are a concern. Individuals cooperate ubiquitously when, for example, environmental protection grows as an issue. Generally, residents living in rural areas raise money to fix water canals and buy cleaning services when public resources or public goods are not on time. These actions helped form repeated or multiple-stage games of participants since individuals are aware of the result of collective benefits greater than the total of individual ones when cooperation is in use.

Corresponding author
Simo Sun, sunsimo@mail.gufe.edu.cn

Hence a synergistic effect occurs. Friedman showed in 1971 that the result of any Nash equilibrium can be constructed in a perfect equilibrium of repeated games (*Friedman, 1971*) when the Pareto condition dominates the original game. In 1976, Aumann and Shapley suggested the replacement of the "Nash Equilibrium" with "Subgame Perfect Equilibrium". The reasons explained by theorists why cooperations emerged and maintained based on varied perspectives. *Robert (1981)* believes that the repeated game of complete information is pertinent to the evolution of the fundamental structure of interactions between agents and proves that cooperation, altruism, revenge, and threat are the outcomes of bounded rationality in real situations. The most proper means to promote cooperation is to employ the robust reciprocity theory (*Rand, Dreber & Elingsen, 2009*) to expect future interests in a long-term repeated game. Cooperation can be formed and maintained by building a reasonably stout reciprocity mechanism for games on any public goods where cooperative equilibrium cannot be formed.

Evolutionary games present a robust theoretical framework to explain how to promote and keep cooperation. However, players may not always maintain the same rationality when strategies are adjusted in real-life situations. Besides, conventional evolutionary game models cannot reflect the effects of uncertain factors such as information opacity and income volatility. *Foster & Young (1990)* proposed the Stochastic Stable Strategy (SSS) concept in 1990 when studying stochastic issues in evolutionary games. Then, many scholars proposed different stochastic evolutionary game models (*Michihiro, George & Rafael, 1993*; *Christine et al., 2004*; *Lorens & Martin, 2006*; *Wallace & Young, 2015*). So far, stochastic games have become a hot research topic for game theorists. In 2018, *Ji et al. (2018)* studied the game model of the random public goods in finite groups. *Li, Zichun & Hui (2020)* considered the impact of random perturbation payment on the equilibrium point in the evolutionary game model.

However, few studies investigate the long-term benefits and stochastic issues in the repeated game of public goods. More up-to-date research regarding reciprocity dynamics in supply chains and evolutionary game theory in the literature (*Ahmad, Shah & Al-Fagih, 2023*; *Hao et al., 2022*). Therefore, the work and innovation of the article introduce a strong reciprocity mechanism in the long-run benefits of the game based on the repeated games of the manufacturer and seller groups in the supply chains of public goods, proposing a model called the evolutionary game to cooperate between these groups in the supply chain of public goods based on the classical evolutionary game models by adding WGN to reflect random interference in the game process (*Sun, Wang & Xue, 2016*; *Li & Xin, 2017*; *Sheng & Chen, 2019*; *Wang & Xia, 2019*). The cooperative evolution process of manufacturer and seller group are analyzed to interpret changes in system stability with random disturbances, and finally examining the effects of different situations on stability through numerical simulations. More up-to-date research regarding the latest research can be found (*Saeedi et al., 2024*; *Ghanei, Contreras & Cordeau, 2023*; *Bilancini et al., 2024*; *Okada, 2023*).

**Table 1  A payoff matrix of a repeated game.**

| | | Seller group | |
| --- | --- | --- | --- |
| | | $C\ (y)$ | $D\ (1-y)$ |
| Manufacturer group | $C\ (x)$ | $\frac{A_1}{1-\overline{\omega}}, \frac{A_2}{1-\overline{\omega}}$ | $\frac{B_1(1-\overline{\omega})-\delta\overline{\omega}}{1-\overline{\omega}}, \frac{(B_2-\delta)(1-\overline{\omega})-\delta\overline{\omega}}{1-\overline{\omega}}$ |
| | $D\ (1-x)$ | $\frac{(C_1-\delta)(1-\overline{\omega})-\delta\overline{\omega}}{1-\overline{\omega}}, \frac{C_2(1-\overline{\omega})-\delta\overline{\omega}}{1-\overline{\omega}}$ | $D_1+\frac{-\delta}{1-\overline{\omega}}, D_2+\frac{-\delta}{1-\overline{\omega}},$ |

# REPEATED GAME EVOLUTION MODEL OF PUBLIC GOODS

## Model assumptions

Scenarios of a game for strategy interactions between the manufacturer group and seller group in the supply chains of public goods are assumed to follow:

1. Both manufacturer and seller groups are affected by the long-term payoff of an infinitely repeated game, which implies that every time the two groups conduct behavioral interactions in a game, there are only two strategies to choose from, cooperation and betrayal, and the two groups have different benefits when they choose different strategies when games are played on public goods.

2. Both groups adopt the "cold trigger" mechanism in the repeated game process. For example, after the first game is run, if the manufacturer group chooses to betray while the seller cooperates in the first round, then the seller group will adopt a ruthless trigger strategy, namely, the betrayal strategy for all. Accordingly, in the second round, the manufacturer group will choose cooperation, and the seller will select defection. The benefits of the seller groups' cooperative strategy and the manufacturers' defection strategy are expressed by the present value of the long-term benefits.

3. Parameter assumption (*Simo et al., 2021*): $\delta(1>\delta>0)$ characterizes the strong reciprocity penalty coefficient, $\overline{\omega}(1 > \overline{\omega} > 0)$ denotes the discount factor, x designates the likelihood that the manufacturer group picks the cooperative (C) strategy, and y shows that of the seller group. D represents the defection strategy. $A_1$, $A_2$, $B_1$, $B_2$, $C_1$, $C_2$, $D_1$, $D_2$ characterize the payoffs of a game between the two groups.

According to the hypothesis, the strong reciprocity mechanism establishes the return matrix of the asymmetric repeated game between the manufacturer and seller groups in the supply chains of public goods. Table 1 presents the mathematical expressions.

## Dynamic system

Equation (1) implies that dynamic systems exist for the manufacturer group and seller group in the supply chains of public goods.

$$\begin{cases} \frac{dx}{dt}=x(1-x)(U_1^C-U_1^D) \\ \frac{dy}{dt}=y(1-y)(U_2^C-U_2^D) \end{cases} \tag{1}$$

where

$$U_1^C = \frac{A_1}{1-\overline{\omega}}y+(1-y)\frac{B_1(1-\overline{\omega})-\delta\overline{\omega}}{1-\overline{\omega}}, \quad U_1^D = \frac{(C_1-\delta)(1-\overline{\omega})-\delta\overline{\omega}}{1-bar\omega}y+(D_1+\frac{-\delta}{1-\overline{\omega}})(1-y);$$
$$U_2^C = \frac{A_2}{1-\overline{\omega}}x+\frac{C_2(1-\overline{\omega})-\delta\overline{\omega}}{1-\overline{\omega}}(1-x), \quad U_2^D = \frac{(B_2-\delta)(1-\overline{\omega})-\delta\overline{\omega}}{1-\overline{\omega}}x+(D_2+\frac{-\delta}{1-\overline{\omega}})(1-x).$$

According to the stable equilibrium of the dynamic system, Eq. (1) has five equilibrium points $(0,0)$, $(0,1)$, $(1,0)$, $(1,1)$ and $(\frac{(D_2-C_2-\delta)(1-\overline{\omega})}{A_2+\delta\overline{\omega}+(D_2-B_2-C_2)(1-\overline{\omega})}, \frac{(D_1-B_1-\delta)(1-\overline{\omega})}{A_1+\delta\overline{\omega}+(D_1-B_1-C_1)(1-\overline{\omega})})$, respectively.

According to the arrow analysis method of the pure strategy of the Nash Equilibrium, when $\frac{A_1}{1-\overline{\omega}}>\frac{(C_1-\delta)(1-\overline{\omega})-\delta\overline{\omega}}{1-\overline{\omega}}$, $\frac{A_2}{1-\overline{\omega}}>\frac{(B_2-\delta)(1-\overline{\omega})-\delta\overline{\omega}}{1-\overline{\omega}}$, namely, $A_1>C_1(1-\overline{\omega})-\delta$, $A_2>B_2(1-\overline{\omega})$ $-\delta$, Eq. (1) has a stable equilibrium point at $(0,0)$, and also an optimal equilibrium point. The outcome shows a stable Eq. (1) strategy when the internal environment is deterministic. Nevertheless, owing to the complexity of public goods' supply chains and the external uncertainty, random factors disturb decision-makers. Therefore, the question "Is the strategy $(0,0)$ still stable with the influence of random factors?" needs to be answered. If it is stable, then what are the conditions? To this end, factors characterized as random disturbance influence stability that needs to be discussed.

The non-negativity of $1-x$, $1-y$ in Eq. (1) does not affect the evolution result of the strategy equilibrium. Therefore, Eq. (1) is changed to the following dynamic system for better discussion.

$$\begin{cases} dx = x(t)([A_1+\delta\overline{\omega}+(D_1-B_1-C_1)(1-\overline{\omega})]y(t)-(D_1-B_1-\delta)(1-\overline{\omega}))dt \\ dy = y(t)([A_2+\delta\overline{\omega}+(D_2-B_2-C_2)(1-\overline{\omega})]x(t)-(D_2-C_2-\delta)(1-\overline{\omega}))dt \end{cases} \quad (2)$$

The stochastic dynamical system after adding WGN into Eq. (2) is defined by

$$dx = x(t)([A_1+\delta\overline{\omega}+(D_1-B_1-C_1)(1-\overline{\omega})]y(t)-(D_1-B_1-\delta)(1-\overline{\omega}))dt + \sigma x(t)dw(t) \quad (3)$$

$$dy = y(t)([A_2+\delta\overline{\omega}+(D_2-B_2-C_2)(1-\overline{\omega})]x(t)-(D_2-C_2-\delta)(1-\overline{\omega}))dt + \sigma y(t)dw(t) \quad (4)$$

where $\sigma$ denotes the random disturbance intensity. According to the group evolution dynamics (*Sheng & Chen, 2019*), let $\sigma[x(t)]=x(t)(1-x(t))$, $w(t)$ denote the standard Brownian motion; $dw(t)$ represents the WGN and follows the normal distribution $N(0,\triangle t)$. Then, the dynamical system represents the strategy transformation of the manufacturer and seller groups with random disturbances, respectively.

## REPEATED RANDOM EVOLUTION MODEL AND STABILITY OF PUBLIC GOODS SUPPLY

According to the analysis in Section 'Dynamic System', when the parameters satisfy $A_1>C_1(1-\overline{\omega})-\delta$, $A_2>B_2(1-\overline{\omega})-\delta$ the conditions, Eq. (1) has a stable equilibrium point at $(0,0)$, but when the system is subject to random disturbance, the question "Is the strategy $(0,0)$ still stable?" needs to be answered. Random disturbance influences stability, so the zero solution stability of stochastic differential equation is conceptualized.

**Definition 1** (*Cobbl, 1985*; *Zhang, Xue & Zhou, 2019*; *Xu, Liu & Qian, 2011*; *Hu, Huang & Wu, 2008*): Let the stochastic process $X=\{X(t), t \geq 0\}$ be the solution to the initial value problem of the following Itô differential equation, $\begin{cases} dX(t)=f(t,X(t))dt+g(t,X(t))dB_t \ \forall t \geq 0 \\ X(t_0)=x_0 \end{cases}$

If there is a negative k-moment Liapunov exponent in $X(t)$, *i.e.*, $\overline{\lim_{t\to\infty}} t^{-1}\ln E|X(t)|^k < 0$, then the equation's solution is said to be stable in terms of the k-expected moment exponent; if $\lim_{t\to\infty} t^{-1}\ln E|X(t)|^k > 0$, then the solution is unstable.

To use Lemma 1 to find the stability of the solution,

**Lemma 1** (*Cobbl, 1985*; *Zhang, Xue & Zhou, 2019*; *Xu, Liu & Qian, 2011*; *Hu, Huang & Wu, 2008*): Let the random process $X = \{X(t), t \geq 0\}$ be the solution of the initial value problem in Eq. (5),

$$\begin{cases} dX(t) = f(t, X(t))\,dt + g(t, X(t))\,dB_t \ \forall t \geq 0 \\ X(t_0) = x_0 \end{cases} \tag{5}$$

where $V(t,x)$ is a continuously differentiable mapping with $c_1, c_2$ positive real numbers that lead to

$$c_1|x|^k \leq V(t,x) \leq c_2|x|^k. \tag{6}$$

(1) If there exists a positive constant $\gamma$ such that $LV(t,x) \leq -\gamma V(t,x)$, the solution of the initial value problem of the Itô differential equation is called exponentially stable in terms of the k-expected moment, *i.e.*, $\overline{\lim_{t\to\infty}} t^{-1}\ln E|X(t)|^k < 0$.

(2) If there is a positive constant $\gamma$ that makes $LV(t,x) \geq \gamma V(t,x)$, the solution of the initial value problem of Itô differential equation is said to be exponentially unstable to the k-expected moment, that is, $\overline{\lim_{t\to\infty}} t^{-1}\ln E|X(t)|^k > 0$.

where $LV(t,x) = V_t(t,x) + V_x(t,x)\,f(t,x) + \frac{1}{2}g^2(t,x)\,V_{(x,x)}(t,x)$.

According to Lemma 1, sufficient conditions for the exponential stability of the k-expected moment of the zero solution in the stochastic system are attained.

**Proposition 1** For the stochastic differential Eq. (3),

$$dx = x(t)([A_1 + \delta\overline{\omega} + (D_1 - B_1 - C_1)(1-\overline{\omega})]y - (D_1 - B_1 - \delta)(1-\overline{\omega}))dt + \sigma x(t)dw(t)$$

There exist $c_1 = c_2 = 1$, $k = 1$, $\gamma = 1$ and Liapunov equation $V(t,x) = x(t)$,

When (1) $A_1 + \delta\overline{\omega} + (D_1 - B_1 - C_1)(1-\overline{\omega}) = 0$, $\delta \leq D_1 - B_1 - \frac{1}{1-\overline{\omega}}$, or

(2) $A_1 + \delta\overline{\omega} + (D_1 - B_1 - C_1)(1-\overline{\omega}) > 0$, $\delta < \min\{D_1 - B_1 - \frac{1}{1-\overline{\omega}}, \frac{C_1 - A_1 - 1}{1 - A_1\overline{\omega} + \overline{\omega} - \overline{\omega}\overline{\omega}}\}$ or

(3) $A_1 + \delta\overline{\omega} + (D_1 - B_1 - C_1)(1-\overline{\omega}) < 0$, $\delta \geq D_1 - B_1 - \frac{1}{1-\overline{\omega}}$, the expected moment of the equation's zero solution is exponentially stable.

When (4) $A_1 + \delta\overline{\omega} + (D_1 - B_1 - C_1)(1-\overline{\omega}) = 0$, $\delta \geq D_1 - B_1 + \frac{1}{1-\overline{\omega}}$, or

(5) $A_1 + \delta\overline{\omega} + (D_1 - B_1 - C_1)(1-\overline{\omega}) > 0$, $\delta \geq D_1 - B_1 + \frac{1}{1-\overline{\omega}}$, or

(6) When $A_1 + \delta\overline{\omega} + (D_1 - B_1 - C_1)(1-\overline{\omega}) < 0$, $\delta > \max\{D_1 - B_1 + \frac{1}{1-\overline{\omega}}, 1 - A_1 + C_1(1-\overline{\omega})\}$, the k-expected moment of the equation's zero solution is exponentially unstable.

Prove: Considering the stochastic differential Eq. (3)

Assume that the Liapunov function $V(t,x) = x(t)$, $c_1 = c_2 = 1$, $k = 1$ exists which satisfies Eq. (5) in lemma 1, and there is $LV(t,x) = f(t,x) = x(t)([A_1 + \delta\overline{\omega} + (D_1 - B_1 - C_1)(1-\overline{\omega})]y(t) - (D_1 - B_1 - \delta)(1-\overline{\omega}))$. According to Lemma 1, if $\gamma = 1$ makes $Lv(t,x) \leq -V(t,x)$, then, the zero-solution k-expected moment of the equation is exponentially stable, so

$LV(t,x) = f(t,x) = x(t)([A_1 + \delta\overline{\omega} + (D_1 - B_1 - C_1)(1-\overline{\omega})]y(t) - (D_1 - B_1 - \delta)(1-\overline{\omega})) \leq -x(t),$

Therefore, $x(t)([A_1 + \delta\overline{\omega} + (D_1 - B_1 - C_1)(1-\overline{\omega})]y(t) - (D_1 - B_1 - \delta)(1-\overline{\omega}) + 1) \leq 0$

As $x(t) \in [0,1]$, so long as

$[A_1 + \delta\overline{\omega} + (D_1 - B_1 - C_1)(1-\overline{\omega})]y(t) - (D_1 - B_1) - \delta)(1-\overline{\omega}) + 1 \leq 0$.

Therefore, the following situations are discussed:

(1) When $A_1 + \delta\overline{\omega} + (D_1 - B_1 - C_1)(1-\overline{\omega}) = 0$, there is $(D_1 - B_1 - \delta))(1-\overline{\omega}) \geq 1$, namely $D_1 - B_1 - \frac{1}{1-\overline{\omega}} \geq \delta$;

(2) When $A_1 + \delta\overline{\omega} + (D_1 - B_1 - C_1)(1-\overline{\omega}) > 0$,

and when $y(t) = 0$ 时, there is $(D_1 - B_1 - \delta)(1-\overline{\omega}) \geq 1$, 即 $\delta \leq D_1 - B_1 - \frac{1}{1-\overline{\omega}}$.

and if $y(t) = 1$, $A_1 + \delta\overline{\omega} + (D_1 - B_1 - C_1)(1-\overline{\omega}) - (D_1 - B_1 - \delta))(1-\overline{\omega}) + 1 \leq 0$, then $\delta(1 - A_1\overline{\omega} + \overline{\omega} - \overline{\omega}\overline{\omega}) + A_1 - C_1 + 1 \leq 0$, namely $\delta \leq \frac{C_1 - A_1 - 1}{1 - A_1\overline{\omega} + \overline{\omega} - \overline{\omega}\overline{\omega}}$.

Therefore: $\delta < \min\{D_1 - B_1 - \frac{1}{1-\overline{\omega}}, \frac{C_1 - A_1 - 1}{1 - A_1\overline{\omega} + \overline{\omega} - \overline{\omega}\overline{\omega}}\}$, $D_1 - B_1 - \frac{1}{1-\overline{\omega}} \geq \delta$.

(3) When $A_1 + \delta\overline{\omega} + (D_1 - B_1 - C_1)(1-\overline{\omega}) < 0$, then, there is $(D_1 - B_1 - \delta)(1-\overline{\omega}) - 1 \leq 0$, $(D_1 - B_1 - \delta)) \leq \frac{1}{1-\overline{\omega}}$, $\delta \geq D_1 - B_1 - \frac{1}{1-\overline{\omega}}$, $(D_1 - B_1 - \delta)(1-\overline{\omega}) - 1 > 0$, that is, $D_1 - B_1 - \delta > (1-\overline{\omega})$.

On the other hand, from lemma 1, if there is $\gamma = 1$, such that $LV(t,x) = f(t,x) \geq x(t)$, the k-expected exponential moment of the initial value problem's solution of Itô differential equation is unstable. It can be obtained from

$LV(t,x) = f(t,x) = x(t)[A_1 + \delta\overline{\omega} + (D_1 - B_1 - C_1)(1-\overline{\omega})]y(t) - (D_1 - B_1 - \delta)(1-\overline{\omega}) \geq x(t)$

that

$x(t)([A_1 + \delta\overline{\omega} + (D_1 - B_1 - C_1)(1-\overline{\omega})]y(t) - (D_1 - B_1 - \delta)(1-\overline{\omega}) - 1) \geq 0$

As $x(t) \in (0,1)$, so long as

$[A_1 + \delta\overline{\omega} + (D_1 - B_1 - C_1)(1-\overline{\omega})]y(t) - (D_1 - B_1 - \delta)(1-\overline{\omega}) - 1 \geq 0$;

Therefore, the following situations are discussed:

(4) When $A_1 + \delta\overline{\omega} + (D_1 - B_1 - C_1)(1-\overline{\omega}) = 0$, $-(D_1 - B_1 - \delta)(1-\overline{\omega}) - 1 \geq 0$, $(D_1 - B_1 - \delta) \leq \frac{-1}{1-\overline{\omega}}$, $\delta \geq D_1 - B_1 + \frac{1}{1-\overline{\omega}}$.

(5) When 当 $A_1 + \delta\overline{\omega} + (D_1 - B_1 - C_1)(1-\overline{\omega}) > 0$, $(D_1 - B_1 - \delta)(1-\overline{\omega}) + 1 \leq 0$, $\delta \geq D_1 - B_1 + \frac{1}{1-\overline{\omega}}$;

(6) When $A_1 + \delta\overline{\omega} + (D_1 - B_1 - C_1)(1-\overline{\omega}) < 0$, and when $y(t) = 0$, there is $(D_1 - B_1 - \delta)(1-\overline{\omega}) + 1 \leq 0$, that is: $\delta \geq D_1 - B_1 + \frac{1}{1-\overline{\omega}}$.

and when $y(t) = 1$, there is $A_1 + \delta\overline{\omega} + (D_1 - B_1 - C_1)(1-\overline{\omega}) - (D_1 - B_1 - \delta)(1-\overline{\omega}) - 1 \geq 0$, and $\delta + A_1 - C_1(1-\overline{\omega}) - 1 \geq 0$, that is: $\delta \geq 1 - A_1 + C_1(1-\overline{\omega})$;

So: $\delta > \max\{D_1 - B_1 + \frac{1}{1-\overline{\omega}}, 1 - A_1 + C_1(1-\overline{\omega})\}$

Proposition 2 is obtained similarly.

**Proposition 2** For the stochastic differential Eq. (4)

For $dy = y(t)([A_2 + \delta\overline{\omega} + (D_2 - B_2 - C_2)(1-\overline{\omega})]x(t) - (D_2 - C_2 - \delta)(1-\overline{\omega}))dt + \sigma y(t)dw(t)$, there are $c_1 = c_2 = 1$, $k = 1$, $\gamma = 1$ and the Liapunov function $V(t,y) = y(t)$, when

(1) $A_2 + \delta\overline{\omega} + (D_2 - B_2 - C_2)(1-\overline{\omega}) = 0$, $\delta \leq D_2 - C_2 - \frac{1}{1-\overline{\omega}}$, or

(2) $A_2 + \delta\overline{\omega} + (D_2 - B_2 - C_2)(1-\overline{\omega}) > 0$, $\delta < \min\{D_2 - C_2 - \frac{1}{1-\overline{\omega}}, A_2 - B_2(1-\overline{\omega}) - 1\}$, or

(3) when $A_2 + \delta\overline{\omega} + (D_2 - B_2 - C_2)(1-\overline{\omega}) < 0$, $\delta \leq D_2 - C_2 - \frac{1}{1-\overline{\omega}}$, the expected moment of the zero solution of the equation is exponentially stable when

(4) $A_2 + \delta\overline{\omega} + (D_2 - B_2 - C_2)(1-\overline{\omega}) = 0$, $\delta \geq D_2 - C_2 + \frac{1}{1-\overline{\omega}}$, or

(5) $A_2 + \delta\overline{\omega} + (D_2 - B_2 - C_2)(1 - \overline{\omega}) > 0$, $\delta \geq D_2 - C_2 + \frac{1}{1-\overline{\omega}}$, or

(6) $A_2 + \delta\overline{\omega} + (D_2 - B_2 - C_2)(1 - \overline{\omega}) < 0$, $\delta > \max\{D_2 - C_2 + \frac{1}{1-\overline{\omega}}, 1 - A_2 + B_2(1-\overline{\omega})\}$, the zero solution of the equation is exponentially unstable with the k moment.

**To prove,** consider the stochastic differential Eq. (4)

Assume that the Liapunov function V(t,y) = y(t), then there is $c_1 = c_2 = 1$, k = 1, which satisfies Eq. (6) in Lemma 2, and there is LV(t,y) = f(t,y) = y(t)([$A_2 + \delta\overline{\omega} + (D_2 - B_2 - C_2)(1-\overline{\omega})$]x(t) − ($D_2 - C_2 - \delta$)(1 − $\overline{\omega}$)). From Lemma 2, if there is $\gamma = 1$, such that Lv(t,y) ≤ − V(t,y), then the zero-solution k-expected moment of the equation is exponentially stable, and therefore, there is

$LV(t,y) = f(t,y) = y(t)([A_2 + \delta\overline{\omega} + (D_2 - B_2 - C_2)(1-\overline{\omega})]x(t) - (D_2 - C_2 - \delta)(1-\overline{\omega})) \leq$
$-y(t)$,

and therefore, $y(t)([A_2 + \delta\overline{\omega} + (D_2 - B_2 - C_2)(1-\overline{\omega})]x(t) - (D_2 - C_2 - \delta)(1-\overline{\omega}) + 1)$
$\leq 0$

As $y(t) \in (0,1)$, so long as

$[A_2 + \delta\overline{\omega} + (D_2 - B_2 - C_2)(1-\overline{\omega})]x(t) - (D_2 - C_2 - \delta)(1-\overline{\omega}) + 1 \leq 0$

Therefore, the following situations are discussed.

(1) When $A_2 + \delta\overline{\omega} + (D_2 - B_2 - C_2)(1 - \overline{\omega}) = 0$, there is $-(D_2 - C_2 - \delta)(1 - \overline{\omega}) + 1 \leq 0$, namely: $D_2 - C_2 - \delta \geq \frac{1}{1-\overline{\omega}}$, then $D_2 - C_2 - \frac{1}{1-\overline{\omega}} \geq \delta$ ;

(2) When $A_2 + \delta\overline{\omega} + (D_2 - B_2 - C_2)(1 - \overline{\omega}) > 0$,

Suppose that x(t) = 0, if there is $-(D_2 - C_2 - \delta)(1-\overline{\omega}) + 1 \leq 0$, then $D_2 - C_2 - \frac{1}{1-\overline{\omega}} \leq 0$.

When x(t) = 1, if there is $A_2 + \delta\overline{\omega} + (D_2 - B_2 - C_2)(1-\overline{\omega}) - (D_2 - C_2 - \delta)(1-\overline{\omega}) + 1$
$\leq 0$, then there is $A_2 - B_2(1-\overline{\omega}) - 1 - \delta \geq 0$, that is: $\delta \leq A_2 - B_2(1-\overline{\omega}) - 1$.

So $\delta \leq \min\{ D_2 - C_2 - \frac{1}{1-\overline{\omega}}, A_2 - B_2(1-\overline{\omega}) - 1\}$

(3) When $A_2 + \delta\overline{\omega} + (D_2 - B_2 - C_2)(1-\overline{\omega}) < 0$, $(D_2 - C_2 - \delta)(1-\overline{\omega}) \geq 1$, namely: $D_2 - C_2 - \frac{1}{1-\overline{\omega}} \geq \delta$

On the other hand, according to lemma 1, if there is $\gamma = 1$, such that $LV(t,x) = f(t,x) \geq x(t)$, the k-expected moment exponent of the initial value problem solution of Itó differential equation is said to be unstable. It can be obtained from

$LV(t,y) = f(t,y) = y(t)([A_2 + \delta\overline{\omega} + (D_2 - B_2 - C_2)(1-\overline{\omega})]x(t) - (D_2 - C_2 - \delta)(1-\overline{\omega})) \geq y(t)$.

That $y(t)([A_2 + \delta\overline{\omega} + (D_2 - B_2 - C_2)(1-\overline{\omega})]x(t) - (D_2 - C_2 - \delta)(1-\overline{\omega})) - 1 \geq 0$

As $y(t) \in (0,1)$, so long as

$[A_2 + \delta\overline{\omega} + (D_2 - B_2 - C_2)(1-\overline{\omega})]x(t) - (D_2 - C_2 - \delta)(1-\overline{\omega}) - 1 \geq 0$

Therefore, the following situations are discussed.

(4) When $A_2 + \delta\overline{\omega} + (D_2 - B_2 - C_2)(1-\overline{\omega}) = 0$, $-(D_2 - C_2 - \delta)(1-\overline{\omega}) - 1 \geq 0$, namely, $D_2 - C_2 + \frac{1}{1-\overline{\omega}} \leq \delta$.

(5) When $A_2 + \delta\overline{\omega} + (D_2 - B_2 - C_2)(1-\overline{\omega}) > 0$, $-(D_2 - C_2 - \delta)(1-\overline{\omega}) - 1 \geq 0$, namely, $D_2 - C_2 + \frac{1}{1-\overline{\omega}} \leq \delta$.

(6) When $A_2 + \delta\overline{\omega} + (D_2 - B_2 - C_2)(1-\overline{\omega}) < 0$,

Let x(t) = 0, and there is $-(D_2 - C_2 - \delta)(1-\overline{\omega}) - 1 \geq 0$, that is: $D_2 - C_2 + \frac{1}{1-\overline{\omega}} \leq \delta$.

When x(t) = 1, and there is $A_2 + \delta\overline{\omega} + (D_2 - B_2 - C_2)(1-\overline{\omega}) - (D_2 - C_2 - \delta)(1-\overline{\omega}) - 1 \geq 0$, so $A_2 - B_2(1-\overline{\omega}) + \delta - 1 \geq 0$, that is: $\delta \geq 1 - A_2 + B_2(1-\overline{\omega})$

and therefore, $\delta \geq \max\{ D_2 - C_2 - \frac{1}{1-\overline{\omega}}, 1 - A_2 + B_2(1-\overline{\omega})\}$.

**Peer**J Computer Science

From Propositions 1 and 2, if the parameters satisfy any one of the exponential stability conditions 1 through 3 of the zero solution k-expected moment, the stochastic dynamic system's equilibrium point (0,0) is constituted of Eqs. (3) and (4), respectively, and is exponentially stable in terms of the k-expected moment, that is, (0,0) is still an evolutionarily stable strategy under random disturbances. Likewise, if the parameters satisfy any one of the conditions of Propositions 4 through 6, the equilibrium point of the system (0,0) for the k-expected moment is exponentially unstable.

## NUMERICAL SIMULATION

To describe intuitively the differential equations' evolution process, the dynamic evolution process is simulated for distinct parameters. From the income matrix, assuming that the incomes of the manufacturer and seller groups in the repeated game of public goods take different parameter values. Then, the numerical simulation of the evolutionary game process is executed using the combination of these different parameters. The following two situations are mainly considered for numerical simulation:

**Scenario 1** In the stable evolution process of cooperative interaction, Set $A_1 = 2$, $\delta = 0.4$, $\overline{\omega} = 0.5$, $D_1 = 3$, $B_1 = 0$, $C_1 = 4$, $A_2 = 4.5$, $D_2 = 3.8$, $B_2 = 2.5$, $C_2 = 1$, then the parameters satisfies

Proposition 1 and Condition 2) $A_1 + \delta\overline{\omega} + (D_1 - B_1 - C_1)(1 - \overline{\omega}) > 0$, $\delta = 0.4 < \min\{D_1 - B_1 - \frac{1}{1-\overline{\omega}}, \frac{C_1 - A_1 - 1}{1 - A_1\overline{\omega} + \overline{\omega} - \overline{\omega}\omega}\} = 1$ and

Proposition 2 and Condition 2) $A_2 + \delta\overline{\omega} + (D_2 - B_2 - C_2)(1 - \overline{\omega}) > 0$, $\delta = 0.4 < \min\{D_2 - C_2 - \frac{1}{1-\overline{\omega}}, A_2 - B_2(1 - \overline{\omega}) - 1\} = 0.8$, 且 $A_1 > C_1(1 - \overline{\omega}) - \delta$, $A_2 > B_2(1 - \overline{\omega}) - \delta$.

Then, the equation's zero-solution k-expected moment is stable exponentially, and the stochastic system will evolve from (1,1) to the stable strategy (0,0).

Numerical simulation 1: Numerical simulation is carried out according to the parameter values in case 1, and the random interference intensity is taken as $\sigma = 0$, 0.25, 0.75, and 2, respectively. The cooperative strategy's evolution process is simulated for the manufacturer and seller groups and is presented in Fig. 1.

1. Figures 1A shows that when there is no random disturbance in the system, the strategies of the manufacturer and seller groups evolve to a stable equilibrium state of cooperative strategy after some time passes. This is mainly because their subjective cognition and information mastery rationally limit the players. Still, manufacturers and sellers have gradually mastered commodity and shopping characteristics and become more rational through market operation. Rational players can only benefit from cooperation.

2. Figures 1B depict that as the system is subject to less random disturbance, the cooperation strategy of the manufacturer and seller groups will fluctuate and then evolve to a stable equilibrium state of cooperation. Due to the players' rational limitations in subjective cognition and information mastery, there will be inconsistent strategic judgments when the system is affected by uncertain and random events, such as emergencies, thus resulting in inconsistent decision-making and volatile cooperation strategy. This also aligns with objective facts since rational players can benefit only through cooperation.

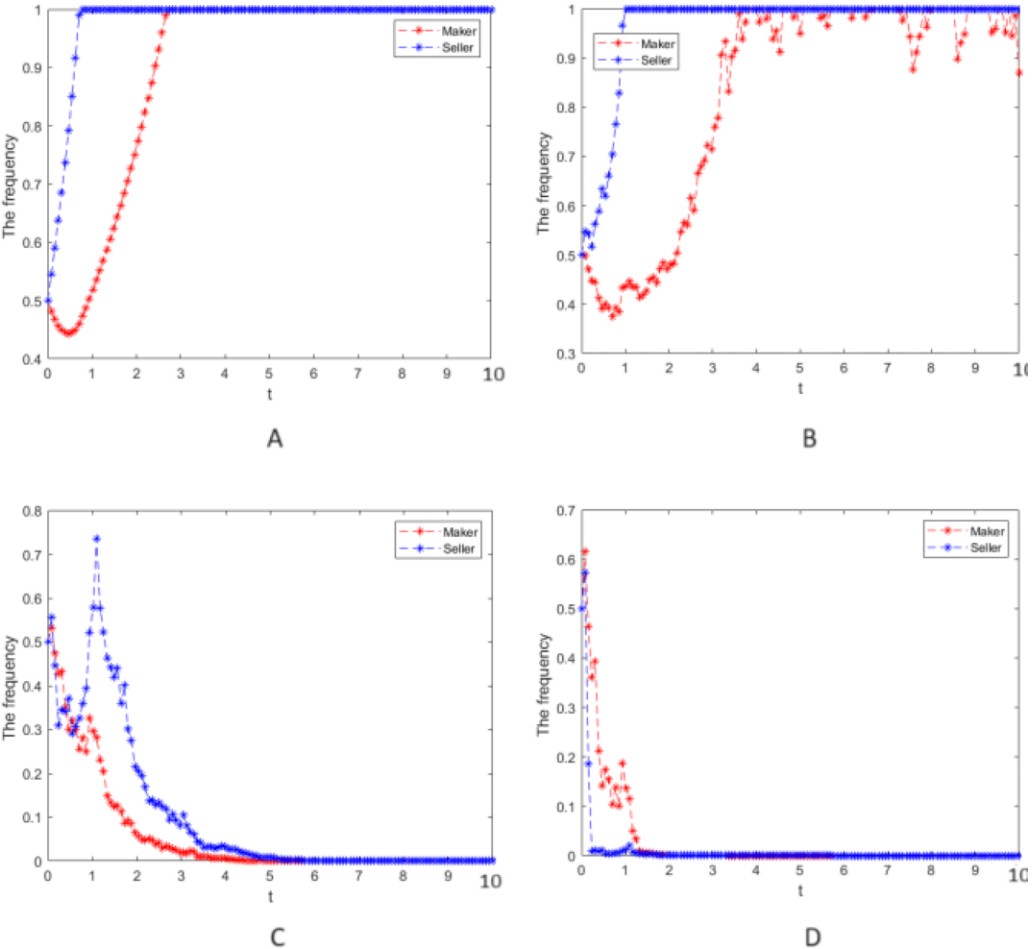

**Figure 1 Numerical simulation of the evolution process.** Numerical simulation of the evolution process of the cooperative strategy of the manufacturer and seller groups.

3. Figures 1C and 1D show that when the random interference is intense, the cooperation strategy of the two groups fluctuates wildly and will deviate from the cooperative strategy (1,1) and evolve to the equilibrium steady state of betrayal policy (0,0). This is because the entirely rational players are subject to greater random interference, making them unable to control their loss of income from random interference such as emergencies. The greater the strength of random interference, the faster the rational players choose a betrayal strategy, and only in this way can the players reduce the losses caused by random interference and obtain the maximum profit.

Numerical simulation 2: In case 1, only the strong reciprocity coefficient, $\delta = 0.6$, in the repeated game is changed, and the values of other parameters and the strength of random interference still take $\sigma = 0$, 0.25, 0.75, and 2, respectively. The cooperative strategy's evolution process is simulated for the manufacturer and the seller groups and can be seen in Fig. 2.

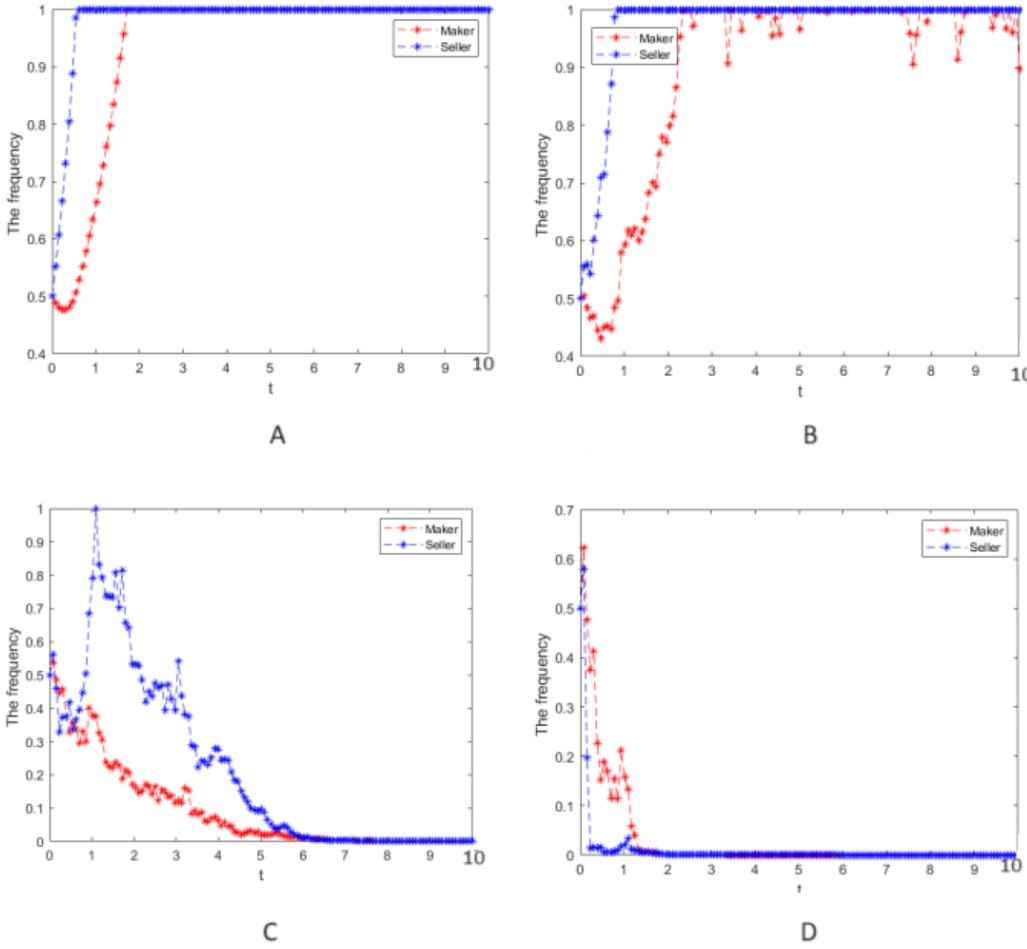

**Figure 2** **Numerical simulation of the evolution process.** Numerical simulation of the evolution process of the cooperative strategy of the manufacturer group and the seller group.

(4) When Figs. 1 and 2 are compared, the larger the strong reciprocity coefficient $\delta$ is, the more the evolution of the cooperation strategy fluctuates with the same random interference intensity, which indicates that the manufacturer group and the seller group pay more attention to the impact from the strong reciprocity coefficient on their earnings when making decisions.

**Scenario 2** The evolution process of cooperative interaction is unstable, set $A_1 = 3$, $\delta = 0.4$, $\overline{\omega} = 0.5$, $D_1 = 1$, $B_1 = 4$, $C_1 = 4.4$, $A_2 = 3$, $D_2 = 1.5$, $B_2 = 4.5$, $C_2 = 5$, then the parameters satisfies

Proposition1 and Condition 6) $A_1 + \delta\overline{\omega} + (D_1 - B_1 - C_1)(1-\overline{\omega}) < 0$, $\delta > \max\{D_1 - B_1 + \frac{1}{1-\overline{\omega}}\}$, $1 - A_1 + C_1(1-\overline{\omega})\}$ and

Proposition 2 and Condition 6) When $A_2 + \delta\overline{\omega} + (D_2 - B_2 - C_2)(1-\overline{\omega}) < 0$, $\delta > \max\{D_2 - C_2 + \frac{1}{1-\overline{\omega}}, 1 - A_2 + B_2(1-\overline{\omega})\}$, 且 $A_1 > C_1(1-\overline{\omega}) - \delta$, $A_2 > B_2(1-\overline{\omega}) - \delta$,

Where the system's zero-solution k-expected moment is unstable exponentially, the random system will not evolve from (1,1) to the stable policy (0,0).

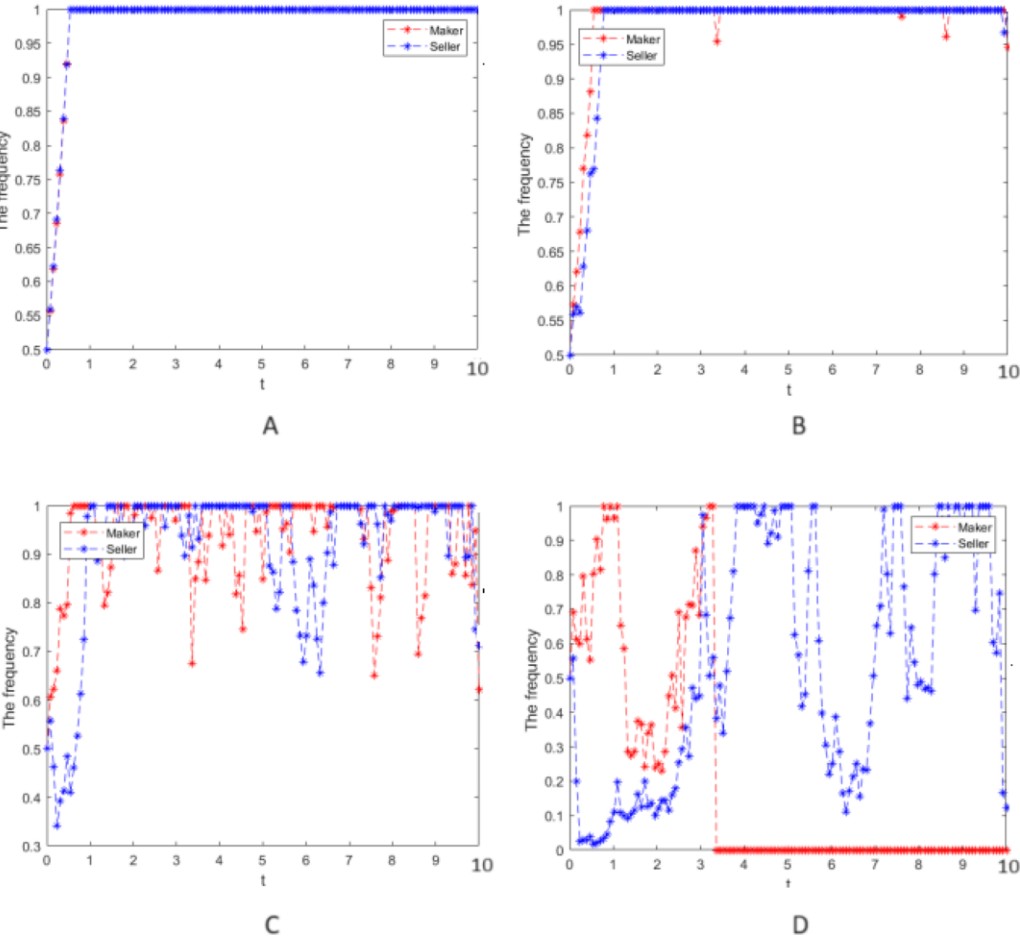

**Figure 3  Numerical simulation of the evolution process.** Numerical simulation of the evolution process of the cooperative strategy of the manufacturer and the seller group.

Numerical simulation 3: Numerical simulation is carried out according to the parameter values in case 2, and the random interference intensity is taken as $\sigma = 0$, 0.25, 0.75, and 2, respectively. The cooperative strategy's evolution process is simulated for the manufacturer and seller groups and is presented in Fig. 3.

(5) Figures 3A and 3B indicate that when there is no or a little random interference in the system, the strategies of the manufacturer and seller groups will soon evolve to a stable equilibrium state of cooperative strategy. This is mainly because rational game players can quickly grasp the market operating rules, and cooperation will result in the maximum benefit.

(6) Figures 3C and 3D show that the larger the random interference intensity, the larger the fluctuation of the cooperation strategy of the two groups would be, and it is unable to evolve from state (1,1) to (0,0). As the parties of the game cannot predict the market environment that randomly occurring factors will impact, the decision-making processes of the manufacturers and seller groups are always swaying.

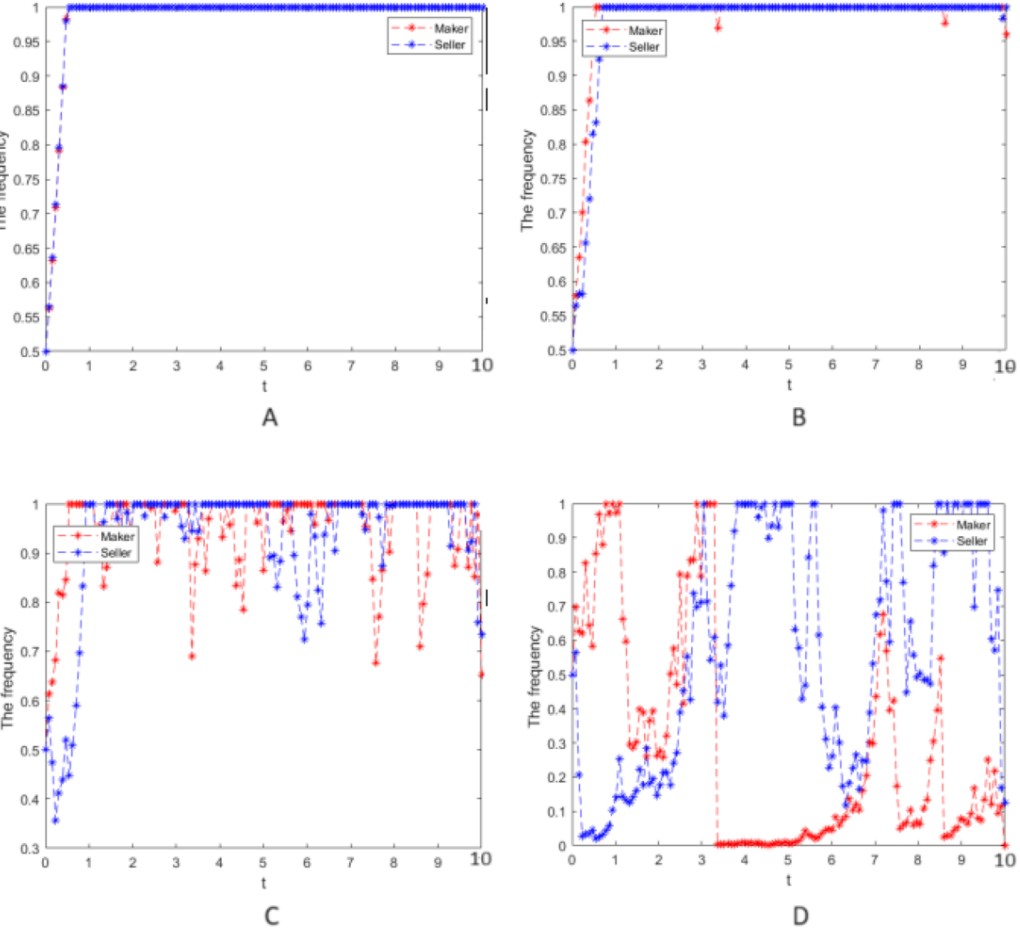

**Figure 4 Numerical simulation of the cooperative strategy evolution.** Numerical simulation of the cooperative strategy evolution process of the two groups.

Numerical simulation 4: In case 3, only the strong reciprocity coefficient $\delta = 0.6$ in the repeated game is changed, and the values of other parameters and the strength of random interference still take $\sigma = 0, 0.25, 0.75, 2$, respectively. The cooperative strategy's evolution process is simulated for the two groups and is depicted in Fig. 4.

(7) When Fig. 3 and 4 are compared, the larger $\delta$, the more significant fluctuation of the cooperation evolution strategy in the case of the same random interference intensity, which means that the two groups pay more attention to the effect of the strong reciprocity coefficient on their earnings when decisions are made, so their hesitation is more prominent.

## CONCLUSION

This article investigates the evolution process of manufacturers' and sellers' behavior strategies in the public goods supply chain. Due to the large period and space in public goods projects and the many factors involved, various uncertainties exist in the operation of public goods. To fully reflect the real state of the system, the manuscript begins with

the repeated game between the manufacturer group and the seller group in supply chains of public goods. It introduces a strong reciprocity mechanism for in-game revenue. It also builds a model called the evolutionary game for the cooperative strategy of the manufacturer and the seller groups by implementing the conventional evolutionary game method. Then, WGN is added to establish a stochastic dynamic system, and the k-moment of the cooperative strategy is simulated to investigate the evolution process.

Research shows that the cooperation strategy of manufacturers and sellers will fluctuate and then stabilize after some time passes when random interference occurs. The players will adopt a cooperative strategy when the strong reciprocity coefficient of betrayal is small. On the other hand, when it is large, the players will adopt a betrayal strategy. Thus, the greater the random interference intensity, the faster it evolves to equilibrium. Namely, the stronger the reciprocity coefficient and the strength of random interference to the system, the greater the changes in selecting strategies when the decision-making processes of players in the repeated game of public goods are under consideration.

According to the research results, the following suggestions are put forward to manage the supply chains of public goods:

(1) To overcome the influence of random factors not directly related to the system but external factors, manufacturers and sellers should pay close attention to the impact of market policies and emergencies and reduce the influence of uncertainties stemming from external factors. Currently, the exchanges and cooperation between manufacturers and sellers need to be strengthened for common information sharing and give benefits to the people to maximize the supply chain's overall revenue in line with the principle of cooperating for a win-win situation.

(2) To make the supply chain of public goods run stably, it is necessary to strengthen the behavior supervision of manufacturers and sellers through the effect of a strong reciprocity policy, to avoid one party choosing a betrayal strategy to maximize individual benefits.

The limitation of the research is that WGN is used to represent randomness. However, other types of randomness could be more realistic in modeling real cases. Instead of using real datasets, we implement simulation data. Even though it is out of the scope of the research, real datasets can provide more reliable results.

### Funding
This research was funded by the Universities Key Laboratory of System Modeling and Data Mining in Guizhou Province of funder grant number No. 2023013 and by the Scientific Research Projects for the Introduced Talents of Guizhou University N0. [2021]90. The funders had no role in study design, data collection and analysis, decision to publish, or preparation of the manuscript.

### Grant Disclosures
The following grant information was disclosed by the authors:

Universities Key Laboratory of System Modeling and Data Mining in Guizhou Province: 2023013.

Scientific Research Projects for the Introduced Talents of Guizhou University: N0. [2021]90.

## Competing Interests

Yi Lei is employed by Guizhou Leading Network Technology Services Co. Ltd. Guizhou Leading Network Technology has no issues to publish this research.

## Author Contributions

- Simo Sun conceived and designed the experiments, performed the computation work, prepared figures and/or tables, authored or reviewed drafts of the article, and approved the final draft.
- Man Wang performed the experiments, performed the computation work, prepared figures and/or tables, authored or reviewed drafts of the article, and approved the final draft.
- Yi Lei analyzed the data, performed the computation work, authored or reviewed drafts of the article, and approved the final draft.

## Data Availability

The raw measurements are available in the Supplementary File.

## Supplemental Information

Supplemental information for this article can be found online at http://dx.doi.org/10.7717/peerj-cs.2118#supplemental-information.

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
