# Peer review of "Analyzing reciprocity dynamics in supply chains of public goods: a stochastic evolutionary game approach"

_PeerJ Computer Science, doi:10.7717/peerj-cs.2118_

## Round 0.1 · original submission · Major Revisions

Dear authors,

Your paper has been read with interest by myself and the expert reviewers. We are of the view that you must incorporate the suggestions of the reviewers and myself given below. Please provide a detailed response and the impact of changes and resubmit.

AE Comments: Could you please elaborate on the process by which the reciprocity mechanism was added to the game income and how it connects to the construction of the dynamic equation? A process flowchart or step-by-step instructions could improve the methodology section's readability.
It would be beneficial to go into more detail about the precise parameters and variables included in the evolutionary game model that is introduced in the research for the cooperative strategy of the seller group and the manufacturer group.

Numerical simulations are mentioned in the paper as a way to examine how various scenarios affect system stability. It would be advantageous to incorporate more thorough insights and conclusions from these simulations

Any study limitations, including assumptions made, oversimplifications of real-world complexity, or limits in the modeling methodology, must be acknowledged

Reviewer 1 ·

Basic reporting

A more thorough discussion of the reciprocity mechanism incorporated into the game income would be beneficial to the study. It would improve the reader's comprehension of the suggested model if this mechanism's operation within the framework of the public goods supply chain game dynamics were explained.

Although the research presents an evolutionary game model for the cooperative strategy of the seller and manufacturer groups, a further explanation of the model's construction would be beneficial. To help readers understand the model's structure and ramifications, it would be helpful to include an explanation of the main variables, parameters, and underlying assumptions.

1. Lack of recent references:
a. No references from the past three years.
b. Only two references from the past five years.

2. Data utilization: You mentioned various datasets available for public goods supply chain, but none were used to test your model ? please justify the reason.

3. Strong reciprocity mechanism: Explain the difference between this mechanism and standard game theory mechanisms, and why you chose it.

4. Random interference: Specify how the random interference reflects real-world uncertainties in the public goods supply chain.

5. Assumptions in model construction: Discuss the assumptions made in building the evolutionary game model and how they might impact the findings.

6. Methodology explanation: Describe in more detail how you analyzed the k-moment of the cooperative strategy and simulated its evolution. What statistical or computational techniques were used?

7. Implications of findings: Discuss how your findings contribute to existing literature on game theory and supply chain management.

8. Limitations and assumptions: Discuss any limitations or assumptions in your study, such as simplifications in modeling or data availability constraints.

9. Addressing criticisms: Address potential criticisms or alternative interpretations of your findings and discuss how robust your conclusions are.

10. Language revision: Consider revising language for clarity and precision, especially in complex or technical sections of the paper.

Experimental design

No comments

Validity of the findings

No Comments

Additional comments

No Comments

Annotated reviews are not available for download in order to protect the identity of reviewers who chose to remain anonymous.

Reviewer 2 ·

Basic reporting

The paper's primary goal, which is to analyze reciprocity dynamics in public goods supply networks using a stochastic evolutionary game technique, is succinctly stated in the title.
It may, however, be made more succinct and understandable by identifying the precise reciprocity dynamics feature that is being examined.

The goal of the study and its methodology are succinctly described in the introduction. Think about giving further background information on the relevance of reciprocity dynamics in supply chains for public goods and the reasons their research is crucial.
To strengthen the theoretical underpinnings of the research, include pertinent publications on reciprocity dynamics in supply chains and evolutionary game theory in the literature review section.
The methodology section offers thorough justifications of the employed technique and is organized with care. But think about going into further detail about why a stochastic evolutionary game approach was chosen and how it varies from other analytical techniques. Provide more clarity on the specific parameters and variables used in the stochastic dynamic system and the differential equation analysis.
The portion on results and debate is well-written. Graphs and charts, on the other hand, would be useful visual aids to better depict the changes in system stability and strategy evolution.
Provide a deeper analysis and interpretation of the numerical results to extract meaningful insights into the relationship between reciprocity coefficients, random interference, and strategy choices.
The main conclusions of the study are outlined in the conclusion.
Think about extending the conclusion to cover the research findings' practical ramifications for public goods supply chains as well as possible directions for further investigation.
Overall, the study uses a stochastic evolutionary game technique to offer insightful information about the dynamics of reciprocity in public goods supply chains. However, the paper's contribution to the area could be strengthened by making more improvements to the contextualization, analysis, and findings presentation.

Experimental design

.

Validity of the findings

.

---

## Round 0.2 · accepted · Accept

Thank you for submitting the revision of your manuscript in light of the comments of the authors and the editor. Based on the input received from the experts, I am please to inform you about the acceptance of your article. Thank you for your fine contribution.

Reviewer 1 ·

Basic reporting

Satisfied

Experimental design

Satisfied

Validity of the findings

Satisfied

Reviewer 2 ·

Basic reporting

This paper has been well revised.

Experimental design

This paper has been well revised.

Validity of the findings

This paper has been well revised.

Additional comments

This paper has been well revised.